# Lignin-Modified Single-Use Graphite Electrodes: Electrochemical Detection of DNA, Mitomycin C, and Their Interaction

**DOI:** 10.3390/s25175427

**Published:** 2025-09-02

**Authors:** Ayla Yıldırım, Meltem Maral, Huseyin Senturk, Arzum Erdem

**Affiliations:** Department of Analytical Chemistry, Faculty of Pharmacy, Ege University, 35040 İzmir, Türkiye

**Keywords:** lignin, mitomycin c, pencil graphite electrode, anticancer drug–DNA interaction, electrochemical DNA biosensor

## Abstract

Lignin, the second most abundant biopolymer in nature after cellulose, has attracted attention for its compatibility with carbon-based materials. In this study, lignin-modified single-use pencil graphite electrodes (PGE/LG) were developed for the electrochemical detection of fish sperm DNA (fsDNA), the anticancer drug Mitomycin C (MC), and their interaction. The modified electrodes were characterized using field emission scanning electron microscopy (FE-SEM), energy-dispersive X-ray spectroscopy (EDX), cyclic voltammetry (CV), and electrochemical impedance spectroscopy (EIS) techniques. Differential pulse voltammetry (DPV) in ferri/ferrocyanide redox probe solution was employed for signal monitoring. The detection limits were calculated as 2.95 ng/mL for fsDNA between 10^1^ and 10^5^ ng/mL and 0.22 pg/mL for MC between 1 and 10^6^ pg/mL. Furthermore, the interaction of DNA with MC was evaluated by DPV and EIS techniques. The cross-linking between MC and the guanine bases of DNA inhibited electron transfer, resulting in a decrease in current response and an increase in charge transfer resistance. These results demonstrate the potential of the PGE/LG platform as a cost-effective, sensitive, and rapid biosensor for DNA detection, anticancer drug analysis, and drug–DNA interaction studies.

## 1. Introduction

Lignin (LG) is an inexpensive, aromatic, amorphous biopolymer that is the second most abundant natural polymer in biomass after cellulose [1]. It is obtained as a by-product of the wood pulp process. In recent years, its applications have been evaluated in various sectors such as electrochemistry, pharmacy, sensors, and biomedicine [2,3]. It shows high affinity for carbon-based materials due to the abundance of aromatic units in its structure. It adsorbs effectively on sp^2^ hybridized carbon surfaces [4]. For this reason, the compatibility of lignin with carbon materials and carbon-based electrodes has been the subject of various studies [3,4,5,6].

DNA analysis plays an important role in various health fields such as disease diagnosis, drug discovery, and forensic medicine. Electrochemical biosensors, in particular, are frequently preferred for DNA detection due to their numerous advantages, including high sensitivity, low detection limits, cost-effectiveness, and ease of use [7].

The determination of drug–DNA interactions using electrochemical DNA biosensors contributes greatly to drug discovery and pharmaceutical development processes [8]. Drug–DNA interactions have been investigated in detail in recent years, especially for chemotherapeutic drugs [9,10,11]. Drug–DNA interactions can occur in various ways, such as covalent bonding/cross-linking, intercalation, DNA cleaving, non-covalent groove binding, and nucleoside–analog coupling [8]. As a result of these interactions, various changes occur in both the drug and DNA molecules, and these changes cause disruptions in the functional properties of DNA [12].

Mitomycin C (MC) is naturally produced by an actinobacterium, *Streptomyces caespitosus* [13]. MC is one of the important chemotherapeutic agents used for the treatment of various types of cancer, including breast, head and neck, esophageal, cervical, stomach, bladder, prostate, and pancreatic cancers [14,15]. MC, which has antitumor and antibiotic activity, has a high side effect profile, and therefore, its widespread use is sometimes limited. Previous studies have shown that MC cross-links with the guanine bases of DNA and thus disrupts DNA synthesis [16,17]. Therefore, careful control of drug quantity and detailed analysis of the interactions between anticancer agents and DNA are essential for the development of new drugs, adjustment of dosages, and evaluation of pharmacokinetic properties. In the literature, Findik et al. (2021) [18] developed a biosensor by modifying pencil graphite electrodes (PGEs) with glycine nanoflowers to investigate MC–DNA interaction. Biosensor responses were evaluated according to guanine oxidation signals obtained by differential pulse voltammetry (DPV). In the linear range of 5 to 20 µg/mL, the detection limit of ctdsDNA was calculated as 1.09 µg/mL. In the linear range of 20 to 100 µg/mL, the MC detection limit was calculated as 12.55 µg/mL. Erdem et al. (2012) [19] developed a biosensor by modifying PGEs with graphene oxide (GO) to investigate MC–DNA interaction. Biosensor responses were evaluated according to guanine oxidation signals obtained by DPV. In the linear range of 20 to 120 µg/mL, the detection limit of dsDNA was calculated as 9.06 µg/mL. In the linear range of 10 to 80 µg/mL, the MC detection limit was calculated as 4.72 µg/mL. In the study conducted by Ensafi et al. [20], an impedimetric DNA biosensor was developed to evaluate the anticancer activity of the chemotherapeutic agent MC on DNA. The biosensor was constructed by immobilizing double-stranded DNA (ds-DNA) onto a pencil graphite electrode (PGE) modified with multi-walled carbon nanotubes (MWCNTs) and poly(diallyldimethylammonium chloride) (PDDA). The interaction between MC and DNA was investigated using electrochemical impedance spectroscopy (EIS). After incubating the DNA-modified electrode in MC solution for a defined time, EIS measurements were then performed, and accordingly, DNA damage was assessed based on changes in the charge transfer resistance (*R_ct_*). The study revealed that native MC did not exhibit significant interaction with DNA. However, upon activation either electrochemically or in acidic conditions, MC demonstrated a marked interaction with DNA. In the study of Karadeniz et al. [21], the interaction between the anticancer drug Mitomycin C (MC) and DNA was investigated electrochemically within a novel drug delivery medium based on a microemulsion system. MC was incorporated into an oil-in-water type microemulsion, and its interaction with DNA was evaluated using DPV in combination with a disposable pencil graphite electrode (PGE). The oxidation signal of guanine on the DNA was monitored before and after interaction with the drug to assess the extent of binding. The physicochemical properties of the microemulsion system were first optimized, and the effects of experimental parameters including MC concentration, incubation time, and DNA concentration were investigated. The results demonstrated that MC interacts with DNA by binding to guanine bases, leading to a significant decrease in the guanine oxidation signal. Furthermore, the study highlights the microemulsion system as a stable and suitable environment for such biosensing applications, offering a promising platform for in vitro monitoring of drug–DNA interactions in the context of anticancer research.

This study aimed to develop lignin-based single-use pencil graphite electrodes for the interaction between fsDNA and anticancer drug MC. The morphological characterization of the developed lignin-modified PGEs was carried out by FE-SEM and EDX; the electrochemical characterization by cyclic voltammetry (CV) and EIS techniques. The lignin-based electrodes were used for the detection of fsDNA and MC by DPV technique based on the difference in redox couple ferri/ferrohexacyanide ([Fe(CN)_6_]^3−/4−^). MC–DNA interaction was studied using various electrochemical techniques such as DPV, EIS, and single frequency impedance (SFI) techniques.

## 2. Materials and Methods

Details of instruments, chemicals, and electrochemical measurements are given in the Appendix A.

### Procedure

The experimental procedure was performed in the steps described below and is shown in Figure 1.

To modify PGE with lignin (LG), 750 µg/mL LG solution was prepared with DMSO, and the solution was sonicated for 60 min.

Preparation of LG-modified PGE: Electrochemical activation of PGE surfaces was carried out by applying + 1.2 V for 30 s in acetate buffer solution (ABS). PGEs were immersed in vials containing 40 µL LG solution for 30 min. Next, the LG-modified PGEs were dried for 15 min. Subsequently, the modified PGEs were activated with 5 mM EDC (1-Ethyl-3-(3-dimethylaminopropyl)carbodiimide)/8 mM NHS (N-Hydroxysuccinimide) for 60 min. All these steps were carried out in the dark. Then the activated electrodes were washed three times with phosphate-buffered saline (PBS).Immobilization of the fsDNA onto LG-modified electrodes: fsDNA was immobilized on the surface of PGE/LG for 30 min. Subsequently, the electrodes were washed three times with ABS.Immobilization of the Mitomycin C (MC) onto LG-modified electrodes: MC was immobilized on the surface of PGE/LG for 30 min in the dark. Subsequently, the electrodes were washed three times with PBS.Interaction of the fsDNA and MC on the electrode surface: fsDNA was immobilized on the surface of PGE/LG for 30 min. Then the electrodes were washed three times with ABS. Subsequently, MC was immobilized on the surface of PGE/LG/fsDNA for 15, 30, and 60 min in the dark. Then the electrodes were washed three times with PBS.

## 3. Results and Discussion

In this study, the optimization and surface characterization of both unmodified pencil graphite electrodes (PGE) and lignin-modified electrodes (PGE/LG) were performed using CV, DPV, EIS, and field emission scanning electron microscopy (FE-SEM) in combination with energy-dispersive X-ray spectroscopy (EDX). CV measurements were evaluated by monitoring changes in the oxidation peak current observed at approximately +0.2 V, recorded in the redox probe solution containing ferri/ferrocyanide ([Fe(CN)_6_]^3−^/^4−^).

Microscopic characterization of PGEs and PGE/LGs was carried out by the FE-SEM technique in two different dimensions, 500 nm and 1 µm (Figure 1). SEM images, despite being obtained at relatively low concentrations, revealed the accumulation of material in the form of irregular and amorphous structures in specific regions of the surface at a concentration of 750 µg/mL. These structures do not exhibit a homogeneous distribution but rather stand out with their blotchy and irregular morphology. The observed non-uniform deposits clearly indicate that the material has interacted with the electrode surface and successfully immobilized. This suggests that the material exhibits strong affinity for the surface even at low concentrations, confirming the effectiveness of the immobilization process. These findings are particularly important as the dense accumulations observed in certain areas reflect a selective adhesion tendency of the material on the electrode surface and provide insight into potential interaction mechanisms [22]. EDX analysis further confirmed the presence of lignin on the electrode surface by revealing an increase in oxygen content from 4.43 wt% for PGE to 5.45 wt% for PGE/LG (Appendix A). This enhancement is attributed to the hydroxyl-rich structure of lignin, suggesting effective surface functionalization through oxygen-containing groups.

LG solutions were prepared using two different solvents (DMF and DMSO) at a concentration of 500 µg/mL and sonicated for 60 min. Graphite pencil electrodes (PGEs) were modified by immersing them in these LG solutions for 30 min, then dried at room temperature for 15 min. Following modification, the electrodes were activated with a covalent coupling agent solution (CA) for 60 min to ensure strong and stable binding of biomolecules to the surface. The EDC in this solution promotes covalent bonding between the carboxyl groups and amine groups on the electrode surface via a carbodiimide group, while NHS increases the efficiency of this reaction, ensuring more effective bonding. As a result, the stability and immobilization efficiency of biomolecules immobilized on the biosensor surface have been significantly increased [23].

The NHS contributes positively to this process by increasing the stability of EDC [24]. To ensure efficient immobilization of DNA on the modified electrode surface, all subsequent experiments were conducted in the presence of the EDC/NHS activation step. Although lignin is inherently a non-conductive material, it contains abundant hydroxyl, methoxy, and phenolic functional groups in its structure [25]. These groups can form strong interactions with biomolecules such as DNA or proteins [26]. These interactions can occur through covalent bonding (e.g., EDC/NHS coupling chemistry) or non-covalent forces such as hydrogen bonds and π–π interactions. This functionally rich surface enhances the efficiency and stability of biomolecule immobilization on the electrode surface. Improved immobilization increases the surface density of the recognition elements, enabling more efficient capture of target molecules and reducing signal loss. Although lignin is inherently non-conductive, its abundant functional groups facilitate strong interactions with biomolecules, thereby enhancing the overall detection sensitivity of the PGE/LG electrode [27,28].

Control experiments were carried out with the same procedure without LG. The influence of solvent type on the electrode modification efficiency was evaluated via CV in the redox probe solution. As shown in Figure 2A and in Appendix A, electrodes modified with LG dissolved in DMF exhibited a 3% increase in oxidation current compared to the unmodified control, whereas those modified with LG dissolved in DMSO showed a significantly higher increase of 17%. Therefore, it was decided to prepare the LG solution in DMSO (283.27 ± 3.69 µA (%RSD, 1.30%, *n* = 3).

To study the effect of LG concentration on surface modification, LG solutions were prepared at six different concentrations, and PGE surfaces were modified. After 250, 500, 750, 1000, 1500, and 3000 µg/mL LG modification onto PGE, there was an increase in average current value by 3%, 17%, 20%, 15%, 14%, and 9%, respectively, compared to the control group. The oxidation current increased progressively with increasing LG concentration, reaching a maximum enhancement of 20% at 750 µg/mL compared to the unmodified control. However, a proportional decrease in current was observed with increasing LG concentration (1000 to 3000 µg/mL), likely due to the formation of a thicker film on the electrode surface that limited electron transfer. Therefore, the optimum LG concentration was determined as 750 µg/mL (Figure 2B, Appendix A).

The effects of modification time (30 and 60 min) (Figure 2C, Appendix A) and CA activation time (5, 30, and 60 min) (Figure 2D, Appendix A) using the determined LG concentration were evaluated. In the average current value compared to the control group, after 30 and 60 min LG modification, there were increases of 20% and 7%, respectively; after 5, 30, and 60 min CA activation, there were increases of 9%, 11%, and 20%, respectively. The highest increase was observed with a 30 min LG modification followed by 60 min of electrode activation compared to the control group and resulting in a current of 291.24 ± 2.03 µA (%RSD, 0.70%, *n* = 3). Based on these findings, the optimum conditions were determined as 750 µg/mL LG modification for 30 min, followed by drying for 15 min, and then electrode activation with CA for 60 min (Appendix A).

The anodic charge transfer values (Q_a_), anodic current values (I_a_), and calculated electroactive surface areas (As) obtained for the PGE control and PGE/LG are given in Appendix A. The electroactive surface areas were calculated by the Randles–Sevcik equation [29] (Equation (1)) to demonstrate the electrode surface modified with LG in contrast to unmodified one:*I_p_* = (2.69 × 10^5^) × *n*^3/2^ × *A* × *D*^1/2^ × *C* × *V*^1/2^,(1)
where *I_p_* = peak current; *n* = number of electrons transferred; *A* = electroactive surface area, cm^2^; *D* = diffusion coefficient, cm^2^/s (i.e, 7.6 × 10^−6^ cm^2^/s); *C* = concentration, mol/cm^3^; and *V* = scan rate, volt/s. [29]

According to the equation, the electroactive surface area of PGE control was 0.292 cm^2^, and the electroactive surface area of PGE/LG was calculated as 0.352 cm^2^. This enhancement can be attributed to the successful surface functionalization via lignin and subsequent EDC/NHS activation, which introduced carboxyl groups facilitating electron exchange with the redox probe solution (Figure 3A). The enhanced current response is thus consistent with a larger effective surface area available for electrochemical reactions.

In addition, the EIS technique has been used for electrochemical characterization of the surfaces of electrodes (Figure 3B). *R_ct_* values obtained with EIS are given in Appendix A.

The average *R_ct_* value was 901.50 ± 78.49 ohm (%RSD, 8.71%, *n* = 2) with PGE control and 489.00 ± 15.56 ohm (%RSD, 3.18%, *n* = 2) with PGE/LG. With PGE/LG, there was a 46% decrease in the average *R_ct_* value in comparison to the control group. This decrease indicates the enhanced electron transfer kinetics at the lignin-modified interface, which is attributed to improved conductivity and surface functionality resulting from the combined effect of lignin and EDC/NHS activation. This is based on the good combination of lignin with the CA agents as a biocompatible material.

The apparent fractional coverage ($\theta_{IS}^{R}$) was calculated according to the equation (Equation (2)) described by Janek et al. [30]:(2)$$\theta_{IS}^{R}=1-\frac{R_{ct}^{a}}{R_{ct}^{b}}$$
where $R_{ct}^{a}$ = the charge transfer resistance before fsDNA immobilization, and $R_{ct}^{b}$ = the charge transfer resistance after fsDNA immobilization.

After immobilization of 0.01 µg/mL fsDNA on PGE control and PGE/LG surfaces, an increase in the average *R_ct_* value of both electrodes was obtained (Figure 3B, Appendix A). However, a higher increase in *R_ct_* value after fsDNA immobilization was observed with PGE/LG compared to PGE control. The apparent fractional coverage was calculated as 0.083 for PGE control and 0.654 for PGE/LG. This effect can be explained by the electrostatic repulsion between the negatively charged phosphate backbone of DNA and the anionic redox probe, which hinders electron transfer at the interface [31].

To support the EIS results, DPV was performed under the same conditions using [Fe(CN)_6_]^3−/4−^ as the redox probe. The peak signal was recorded at approximately +0.2 V (Figure 3C and Appendix A). An amount of 0.01 µg/mL fsDNA was immobilized on both PGE control and PGE/LG surfaces. Following fsDNA immobilization, the PGE/LG showed a 10% decrease in oxidation current, yielding a value of 279.06 ± 5.03 µA (%RSD = 1.80%, *n* = 3), compared to the unmodified PGE. This decrease in signal is consistent with the increased *R_ct_* observed via EIS, reflecting the insulating nature of the DNA layer on the electrode surface and confirming successful DNA immobilization.

The electrochemical performance of the biosensor under optimized conditions was evaluated according to the difference in redox probe signals by DPV. To evaluate the DNA immobilization time, 20 µg/mL fsDNA was immobilized on the PGE/LG surface for 15, 30, and 60 min. The signals were recorded at +0.25 V peak potential. The highest decrease of 46%, resulting in a current of 151.57 ± 10.09 µA (%RSD, 6.66%, *n* = 3), was observed after 30 min of immobilization (Figure 3D, Appendix A). The optimum immobilization time for fsDNA binding was determined to be 30 min.

Following this, the biosensor’s response was investigated across a range of fsDNA concentrations (from 10^1^ to 10^6^ ng/mL) immobilized on the PGE/LG surface (Figure 4A). A consistent decrease in current was observed up to 10^5^ ng/mL, indicating that the biosensor response is dependent on the DNA concentration immobilized on the electrode surface (Appendix A). However, at concentrations exceeding this threshold, the signal plateaued, suggesting surface saturation and limited additional binding capacity (Appendix A). The limit of detection (LOD) was calculated according to the IUPAC method [32] in the linear range 10^1^–10^5^ ng/mL using the calibration plot shown in Figure 4B, and the equation I(µA) = −40.16 log (C_fsDNA_, ng/mL) + 318.20, and was found to be 2.95 ng/mL with a correlation coefficient of 0.9982 (detailed calculation process [33] in the Appendix A). The sensitivity of the developed DNA biosensor was calculated by dividing the slope of the calibration plot (40.16) by the active surface area of PGE/LG (0.352 cm^2^) and was found to be 114.1023 μA ng^−1^ mL cm^−2^.

The anticancer drug Mitomycin C (MC) was selected as a model compound to investigate DNA–drug interactions.

To evaluate the electrochemical behavior of MC immobilization on the lignin-modified electrode (PGE/LG), various concentrations of MC (10^0^–10^7^ pg/mL) were incubated on the electrode surface for 30 min. DPV measurements were recorded at a peak potential of +0.18 V (Figure 4C). A concentration-dependent decrease in current was observed up to 10^6^ pg/mL, indicating effective interaction between MC and the electrode surface. At concentrations above this level, a deviation from linearity was noted, likely due to surface saturation effects (Appendix A, Appendix A). The limit of detection (LOD) was calculated according to the IUPAC method [32] in the linear range 10^0^–10^6^ pg/mL using the calibration plot shown in Figure 4D, and the equation I(µA) = −24.86 log (C_MC_, pg/mL) + 282.42, and was found to be 0.22 pg/mL with a correlation coefficient of 0.9937. The sensitivity of the developed MC biosensor was calculated by dividing the slope of the calibration plot (24.86) by the active surface area of the PGE/LG (0.352 cm^2^) and was found to be 70.6108 μA pg^−1^ mL cm^−2^.

To assess the interaction between MC and DNA, the PGE/LG was sequentially modified with 20 µg/mL fsDNA followed by 0.001 µg/mL MC. Interaction times of 15, 30, and 60 min were evaluated using both DPV and EIS techniques. According to the results obtained by DPV, a decrease in redox probe signals was obtained at each of the different interaction times. The average redox probe signal after fsDNA immobilization on PGE/LG was measured as 151.57 ± 10.09 µA (%RSD, 6.66%, *n* = 3). The highest decrease of 33% (101.71 ± 1.81 µA (%RSD, 1.78%, *n* = 3) in the signals was obtained after 30 min MC interaction (Figure 5A, Appendix A). According to the results by EIS, an increase in *R_ct_* values was obtained at each of the different interaction times. The average *R_ct_* value obtained after fsDNA immobilization on the PGE/LG was 1625.00 ± 35.36 ohm (%RSD, 2.18%, *n* = 2). The highest increase in average *R_ct_* value with 61% (2615.00 ± 63.64 ohm, %RSD, 2.43%, *n* = 2) was obtained after 30 min MC interaction (Figure 5B, Appendix A). DPV and EIS results show a good correlation. The observed reduction in electron transfer, as reflected by both decreased current and increased *R_ct_*, is attributed to the covalent cross-linking of MC to guanine bases within the DNA. This interaction likely forms a dense molecular layer on the electrode, hindering redox probe accessibility and thus suppressing electrochemical responses.

The signal change percentage (*S*%) values were calculated for 30 min interaction time according to the equation [34], which is the ratio of the peak height of the redox probe signal after (*Ss*) and before (*Sb*) interaction as in the following equation (Equation (3)):(3)$$S\%=\left(\frac{S_{s}}{S_{b}}\right)\times 100$$

The DPV signal of the sensor analyte-free is considered “blank” or 100%. A sample is considered toxic if *S* < 50% and non-toxic if *S* > 85%. If the *S* value is 50–85%, the sample is considered moderately toxic. As in the studies in the literature [18,19], the toxic effect of Mitomycin C (MC) on DNA was assessed by examining changes in electrochemical redox peak height values before and after the interaction between MC and DNA. In this approach, the decrease in redox peak height is associated with damage or conformational changes in the DNA structure, thus indirectly determining MC binding to DNA and its toxicity. In our study, the *S* value calculated from the electrochemical signal obtained after incubating MC with fsDNA for 30 min was found to be 67%. The *S* value, generally defined as the proportional expression of the change in peak height, quantitatively reflects the degree of MC binding to DNA and its associated toxic effect. The 67% value obtained indicates that MC has a moderate toxic effect on fsDNA, i.e., it causes a significant but not excessive DNA structural change. These findings support that the DNA binding mechanism of MC can be sensitively monitored by electrochemical methods and that biosensors can be used effectively in toxicity assessment. These results are consistent with previous studies reported in the literature, further validating the reliability of our findings.

Single frequency impedance (SFI) is an electrochemical technique that allows simultaneous monitoring of changes in phase angle and impedance over time at a fixed frequency. This technique allows the study of various biomolecular interactions such as antigen–antibody, aptamer–protein, and drug–DNA interactions occurring on the electrode surface [35,36].

In the study, single-frequency impedance (SFI) analysis was employed to monitor real-time changes in impedance and phase angle during MC–DNA interaction on the PGE/LG surface. SFI measurements were performed at a fixed frequency of 3 Hz for 30 min. As shown in Figure 5C, a significant increase in impedance (Δ*R_ct_* = 336.89 Ω) and a change in phase angle (Δθ = 1.36°) were observed, confirming the occurrence of molecular interaction on the electrode surface. These results, when considered alongside the DPV and EIS data, strongly validate the formation of an MC–DNA complex and highlight the utility of SFI as a complementary electrochemical technique for biosensing applications.

The storage stability of the developed biosensor was evaluated by storing lignin-modified PGEs in the dark at 4 °C for up to 60 days with CV technique. Stability measurements were performed under optimized conditions, excluding EDC/NHS activation, to mimic pre-functionalization storage. The percentage decrease in the average current values (*n* = 3) at the end of days 1, 15, 21, 30, 45, and 60 was 3%, 12%, 12%, 10%, 11%, and 11%, respectively (Appendix A). These findings indicate that the lignin-based sensor platform maintains electrochemical stability over extended storage periods, making it suitable for practical applications.

## 4. Conclusions

In this study, lignin-modified single-use pencil graphite electrodes (PGE/LGs) were developed and employed for the first time as an electrochemical platform for monitoring interactions between a chemotherapeutic agent and DNA. The use of lignin, a sustainable and biocompatible material, enabled the fabrication of cost-effective, sensitive, and stable biosensor surfaces.

Under optimum conditions, the detection limit of fsDNA was calculated as 2.95 ng/mL in the concentration range 10^1^–10^5^ ng/mL, and the detection limit of MC was calculated as 0.22 pg/mL in the concentration range 10^0^–10^6^ pg/mL by the DPV technique. While the responses of biosensors developed in previous studies were evaluated based on oxidation signals, in this study, they were assessed using the redox probe signal. This approach enabled the detection of both DNA and Mitomycin C (MC) at lower concentrations compared to those reported in earlier studies [18,19,20]. The interaction between MC and DNA was also investigated using various techniques such as DPV, EIS, and SFI.

The proposed lignin-based sensor platform offers a promising tool for applications in pharmaceutical research, such as drug–DNA interaction studies, genotoxicity screening, and nucleic acid detection. Moreover, the methodology can be adapted for the detection of a broad range of biomolecular interactions by tailoring the surface chemistry. These findings highlight the potential of lignin-functionalized electrodes in developing next-generation biosensors for biomedical and environmental monitoring applications.

## Data Availability

The data presented in this study are available within the article and its Appendix A. Other data that support the findings of this study are available upon request from the corresponding author as well as co-author.

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
