# Peer review of "Lignin-Modified Single-Use Graphite Electrodes: Electrochemical Detection of DNA, Mitomycin C, and Their Interaction"

_sensors, 2025, doi:10.3390/s25175427_

Round 1
Reviewer 1 Report
Comments and Suggestions for Authors
COMMENTS TO AUTHORS:
In this study, researchers designed a lignin modified disposable pencil graphite electrodes (PGE/LG) electrode for electrochemical detection. The application of lignin in this work provides cost-effective, sensitive, and stable design strategy for electrochemical platforms, which is interesting and valuable. However, this work still requires major improvements before it can be considered for publication. Some comments are given below.
- All the figures in the manuscript are blurry. It might be better to try to replace them with a higher-resolution one.
- In figure 1, the author compared the SEM images of PEG and PGE/LG, it might be better if the author provides the morphology of lignin.
- In Figure 2-5, the author should name the “a,b,c….” in each figure to let readers know the corresponding condition for each line.
- Does the author think the loading dosage of lignin has significant effect on the sensitivity of the composite electrode?
- Lignin is non-conductive material, why the introduction of lignin into PEG could help to enhance the sensitivity of the electrode? It might be better to provide a detailed discussion on the mechanism of this designed electrode.
Author Response
Journal: Sensors
Manuscript ID: sensors-3801114
Title: Lignin modified single-use graphite electrodes: Electrochemical detection of DNA, Mitomycin C and their interaction
Corresponding Author: Arzum Erdem*
Author(s): Ayla Yıldırım, Meltem Maral, Huseyin Senturk, Arzum Erdem *
Received: 21 Jul 2025
August 24, 2025
The list of our answers to the comments of Editor:
Thank you for valuable comments of Editor and Reviewers. Manuscript is revised/corrected according to each comment pointed by Reviewers. The revised/corrected parts in the manuscript are shown as highlighted in red.
Reviewer 1
Open Review
(x) I would not like to sign my review report
( ) I would like to sign my review report
Quality of English Language
( ) The English could be improved to more clearly express the research.
(x) The English is fine and does not require any improvement.
Yes |
Can be improved |
Must be improved |
Not applicable |
|
Does the introduction provide sufficient background and include all relevant references? |
(x) |
( ) |
( ) |
( ) |
Is the research design appropriate? |
(x) |
( ) |
( ) |
( ) |
Are the methods adequately described? |
( ) |
(x) |
( ) |
( ) |
Are the results clearly presented? |
( ) |
(x) |
( ) |
( ) |
Are the conclusions supported by the results? |
( ) |
(x) |
( ) |
( ) |
Are all figures and tables clear and well-presented? |
( ) |
( ) |
(x) |
( ) |
Comments and Suggestions for Authors
COMMENTS TO AUTHORS:
In this study, researchers designed a lignin modified disposable pencil graphite electrodes (PGE/LG) electrode for electrochemical detection. The application of lignin in this work provides cost-effective, sensitive, and stable design strategy for electrochemical platforms, which is interesting and valuable. However, this work still requires major improvements before it can be considered for publication. Some comments are given below.
Answer: First of all, we would like to thank the Reviewer-1 for valuable comments.
- All the figures in the manuscript are blurry. It might be better to try to replace them with a higher-resolution one.
Answer: Thank you for your comment.
All figures in the main text (MS) and supplementary information (Supporting Information) Word files of the current article have been edited and placed in high resolution.
- In figure 1, the author compared the SEM images of PEG and PGE/LG, it might be better if the author provides the morphology of lignin.
Answer: Thank you for your comment.
The relevant image showing the morphology of lignin (LG) is presented in Scheme 1. Therefore, only the SEM images of PEG and PGE/LG are compared in Figure 1. The relevant section can be found below:
Scheme 1. Schematic representation of the experimental protocol for the detection of MC, fsDNA and MC-DNA interaction. The FE-SEM image of the lignin (LG) is provided in the inset.
- In Figure 2-5, the author should name the “a,b,c….” in each figure to let readers know the corresponding condition for each line.
Answer: Thank you for your comment.
In Figures 2–5, each figure subtitle is numbered ‘A, B, C...’, and each data point is labelled with letters such as ‘a, b, c...’. These letters are indicated in the figure captions along with the conditions they correspond to. The relevant section can be found below:
‘’Figure 2. (A) Voltammograms obtained by CV in redox probe solution using DMF as solvent with (a) PGE control, (b) PGE/LG; using DMSO as solvent with (c) PGE control, (d) PGE/LG. (B) Voltammograms obtained by CV in redox probe solution after modification at increasing LG concentrations in the range 250-3000 µg/mL. (a) PGE control and (b) 250, (c) 500, (d) 750, (e) 1000, (f) 1500, and (g) 3000 µg/mL LG modifications onto PGE. Inset fig.: Zoomed version of the image showing oxidation signals measured at 0.27 V. (C) Voltammograms obtained by CV in redox probe solution with LG modification at different times. PGE control for (a) 30 min, and (b) 60 min; PGE/LG for (c) 30 min, and (d) 60 min. (D) Voltammograms obtained by CV in redox probe solution with CA activation at different times. PGE control for (a) 5 min, (b) 30 min, and (c) 60 min; PGE/LG for (d) 5 min, (e) 30 min, and (f) 60 min. Inset fig.: Zoomed version of the image showing oxidation signals measured at 0.27 V.’’
‘’Figure 3. (A) Voltammograms obtained by CV technique; (a) PGE control, and (b) PGE/LG in 0.01 M KCL solution; and the voltammograms of (c) PGE control, and (d) PGE/LG in redox probe solution. (B) Nyquist plots obtained by EIS in redox probe solution with (a) PGE control, (b) PGE/LG and after immobilization of 0.01 µg/mL fsDNA on (c) PGE control, (d) PGE/LG. (C)Voltammograms obtained by DPV in redox probe solution with (a) PGE control, (b) PGE/LG and after immobilization of 0.01 µg/mL fsDNA on (c) PGE control, (d) PGE/LG. (D) Voltammograms obtained by DPV in redox probe solution with (a) PGE/LG and after immobilization of 0.01 µg/mL fsDNA on PGE/LG for (b) 15 min, (c) 30 min, and (d) 60 min.’’
‘’Figure 4. (A) DPVs and (B) calibration plot presenting the average redox probe signal measured (a) PGE/LG and after 30 min immobilization of (b) 101, (c) 102, (d) 103, (e) 104, and (f) 105 ng/mL fsDNA onto the PGE/LG. (C) DPVs and (D) calibration plot presenting the average redox probe signal measured (a) PGE/LG and after 30 min immobilization of (b) 100 (c) 101, (d) 102, (e) 103, (f) 104, (g) 105, and (h) 106 pg/mL MC onto the PGE/LG.’’
‘’Figure 5. (A) Voltammograms obtained by DPV in redox probe solution with (a) PGE/LG (b) PGE/LG/fsDNA and after 0.001 µg/mL MC immobilization for (c) 15 min, (d) 30 min, (e) 60 min on PGE/LG/fsDNA. (B) Nyquist plots obtained by EIS in redox probe solution with (a) PGE/LG (b) PGE/LG/fsDNA and after 0.001 µg/mL MC immobilization for (c) 15 min, (d) 30 min, (e) 60 min on PGE/LG/fsDNA. (C) SFI measurement of MC and DNA interaction at a frequency of 3 Hz.’’
- Does the author think the loading dosage of lignin has significant effect on the sensitivity of the composite electrode?
Answer: Thank you for your comment.
In our study, the effect of different lignin (LG) loading doses on the sensitivity of the composite electrode was investigated. The results obtained show that the amount of lignin has a direct effect on the electrode surface morphology and conductivity. At low lignin dosages (250 µg/mL), sensitivity was limited due to insufficient functional groups on the electrode surface. At excessive lignin loadings (750 µg/mL and above), electrical conductivity decreased and the signal ratio declined. The optimal condition was achieved at a lignin loading of 750 µg/mL, where the sensitivity of the DNA and MC biosensors was found to be 114.1023 µA·ng⁻¹·cm⁻² and 70.6108 µA·pg⁻¹·cm⁻², respectively. This section has been revised, and the relevant section has been revised as follows:
‘’To study the effect of LG concentration on surface modification, LG solutions were prepared at six different concentrations, and PGE surfaces were modified. After 250, 500, 750, 1000, 1500, and 3000 µg/mL LG modification onto PGE, there was an increase in average current value by 3%, 17%, 20%, 15%, 14%, and 9%, respectively, compared to the control group. The oxidation current increased progressively with increasing LG concentration, reaching a maximum enhancement of 20% at 750 µg/mL compared to the unmodified control. In addition, a proportional decrease in current was observed with increasing LG concentration (1000 to 3000 µg/mL), likely due to the formation of a thicker film on the electrode surface that limited electron transfer. Therefore, the optimum LG concentration was determined as 750 µg/mL (Figure 2B, Table S2).”
‘’The limit of detection (LOD) was calculated according to the IUPAC method [28] in the linear range 101- 105 ng/mL using the calibration plot shown in Figure 4B, and the equation I(µA)=-40.16 log (CfsDNA, ng/mL) + 318.20, was found to be 2.95 ng/mL with a correlation coefficient of 0.9982 (Detailed calculation process in the Supplementary Material). The sensitivity of the developed DNA biosensor was calculated by dividing the slope of the calibration plot (40.16) by the active surface area of PGE/LG (0.352 cm2) and was found to be 114.1023 μA ng−1 mL cm−2.’’
‘’The limit of detection (LOD) was calculated according to the IUPAC method [28] in the linear range 100-106 pg/mL using the calibration plot shown in Figure 4D, and the equation I(µA)=-24.86 log (CMC, pg/mL) + 282.42, was found to be 0.22 pg/mL with a correlation coefficient of 0.9937. The sensitivity of the developed MC biosensor was calculated by dividing the slope of the calibration plot (24.86) by the active surface area of PGE/LG (0.352 cm2) and was found to be 70.6108 μA pg−1 mL cm−2.’’
- Lignin is non-conductive material, why the introduction of lignin into PEG could help to enhance the sensitivity of the electrode? It might be better to provide a detailed discussion on the mechanism of this designed electrode.
Answer: Thank you for your comment.
‘’The NHS contributes positively to this process by increasing the stability of EDC [24]. To ensure efficient immobilization of DNA on the modified electrode surface, all subsequent experiments were conducted in the presence of the EDC/NHS activation step. Although lignin is inherently a non-conductive material, it contains abundant hydroxyl, methoxy, and phenolic functional groups in its structure [25]. These groups can form strong interactions with biomolecules such as DNA or proteins [26]. These interactions can occur through covalent bonding (e.g., EDC/NHS coupling chemistry) or non-covalent forces such as hydrogen bonds and π–π interactions. This functionally rich surface enhances the efficiency and stability of biomolecule immobilization on the electrode surface. Improved immobilization increases the surface density of the recognition elements, enabling more efficient capture of target molecules and reducing sig-nal loss. Despite of the lack of conductivity due to lignin immobilization, the overall detection sensitivity of the PEG/LG composite electrode is enhanced [27,28].’’
Added references:
- Laurichesse, S.; Avérous, L. Chemical Modification of Lignins: Towards Biobased Polymers. Prog. Polym. Sci. 2014, 39, 1266–1290.
- Zhao, J.; Zhu, M.; Jin, W.; Zhang, J.; Fan, G.; Feng, Y.; Li, Z.; Wang, S.; Lee, J.S.; Luan, G.; et al. A Comprehensive Review of Unlocking the Potential of Lignin-Derived Biomaterials: From Lignin Structure to Biomedical Application. J. Nanobiotechnology 2025, 23.
- Tunca, N.; Maral, M.; Yildiz, E.; Sengel, S.B.; Erdem, A. Synthesis and Characterization of Polysaccharide-Cryogel and Its Application to the Electrochemical Detection of DNA. Microchim. Acta 2024, 191, doi:10.1007/s00604-024-06550-7.
- Tortolini, C.; Capecchi, E.; Tasca, F.; Pofi, R.; Venneri, M.A.; Saladino, R.; Antiochia, R. Novel Nanoarchitectures Based on Lignin Nanoparticles for Electrochemical Eco‐friendly Biosensing Development. Nanomaterials 2021, 11, 1–17, doi:10.3390/nano11030718.
In addition, apparent fractional coverage () calculations based on EIS (Electrochemical Impedance Spectroscopy) data were performed to evaluate the binding capacity of DNA on the electrode surfaces. The results demonstrated that DNA binding to the electrode surface increased after lignin modification, indicating the advantageous nature of this modification. The relevant section can be found below:
‘’In addition, the EIS technique has been used for electrochemical characterization of the surfaces of electrodes (Figure 3B). Rct values obtained with EIS are given in Table S7.
The average Rct value was 901.50 ± 78.49 ohm (%RSD, 8.71%, n = 2) with PGE control and 489.00 ± 15.56 ohm (%RSD, 3.18%, n = 2) with PGE/LG. With PGE/LG, there was a 46% decrease in the average Rct value compared to the control group. This decrease indicates the enhanced electron transfer kinetics at the lignin-modified interface, which is attributed to improved conductivity and surface functionality resulting from the combined effect of lignin and EDC/NHS activation. This is based on the good combination of lignin and CA, a biocompatible material.
The apparent fractional coverage () was calculated according to the equation (Eq. 2) described by Janek et al. [30];
|
(2) |
where = the charge transfer resistance before fsDNA immobilization, = the charge transfer resistance after fsDNA immobilization.
Submission Date
21 July 2025
Date of this review
25 Jul 2025 00:44:38
Reviewer 2 Report
Comments and Suggestions for Authors
This article is devoted to the production of graphite electrodes modified with lingin for biosensors, which will be used in detecting the interaction of DNA with the anti-cancer drug Mitomycin C. The relevance of the study lies in the need to effectively detect small concentrations of anti-cancer drugs in the maximum permissible concentrations, since exceeding their number can lead to negative consequences for DNA. Therefore, it is extremely important to record the fact of interaction. Using a set of analysis methods for measuring electrical properties, it was found that the electrodes were successfully modified with lingin and the biosensor detected the presence of Mitomycin C in fish DNA. The responses of biosensors developed in previous studies were evaluated based on oxidation signals, in this study, they were assessed using the redox probe signal. This approach enabled the detection of both DNA and mitomycin C at lower concentrations compared to those reported in earlier studies.
There are the following comments on the article:
- In Figure 1, it is necessary to indicate the areas of lingin modification or to indicate changes in the characteristic size of the graphite flakes. In the above form, the effect of the modification on the electron microscope images is weakly noticeable.
- On line 69-70, Mitomycin C is abbreviated by the authors as MMC, although on line 49 it is designated as MS. Subsequently, both abbreviations are present in the text, which leads to misunderstanding. It is necessary either to clarify the difference between these abbreviations, or to correct typos.
- Why did the authors choose such a diffusion coefficient on line 205? It is necessary to add an extended description or an appropriate reference link to the source from which the number was borrowed.
- Lines 322-325 indicate the toxicity of the sample. It is desirable to expand the use of this term, since the analysis of values is based on equation (3). It describes the height of peaks before and after the redox reaction. That is, it is not entirely clear whether they are talking about the toxicity of Mitomycin C for DNA, or by toxicity the authors mean the degree of contamination of the probe.
Author Response
Journal: Sensors
Manuscript ID: sensors-3801114
Title: Lignin modified single-use graphite electrodes: Electrochemical detection of DNA, Mitomycin C and their interaction
Corresponding Author: Arzum Erdem*
Author(s): Ayla Yıldırım, Meltem Maral, Huseyin Senturk, Arzum Erdem *
Received: 21 Jul 2025
August 23, 2025
The list of our answers to the comments of Editor:
Thank you for valuable comments of Editor and Reviewers. Manuscript is revised/corrected according to each comment pointed by Reviewers. The revised/corrected parts in the manuscript are shown as highlighted in red.
Reviewer 2
Open Review
(x) I would not like to sign my review report
( ) I would like to sign my review report
Quality of English Language
( ) The English could be improved to more clearly express the research.
(x) The English is fine and does not require any improvement.
Yes |
Can be improved |
Must be improved |
Not applicable |
|
Does the introduction provide sufficient background and include all relevant references? |
(x) |
( ) |
( ) |
( ) |
Is the research design appropriate? |
( ) |
(x) |
( ) |
( ) |
Are the methods adequately described? |
( ) |
(x) |
( ) |
( ) |
Are the results clearly presented? |
(x) |
( ) |
( ) |
( ) |
Are the conclusions supported by the results? |
(x) |
( ) |
( ) |
( ) |
Are all figures and tables clear and well-presented? |
( ) |
(x) |
( ) |
( ) |
Comments and Suggestions for Authors
This article is devoted to the production of graphite electrodes modified with lingin for biosensors, which will be used in detecting the interaction of DNA with the anti-cancer drug Mitomycin C. The relevance of the study lies in the need to effectively detect small concentrations of anti-cancer drugs in the maximum permissible concentrations, since exceeding their number can lead to negative consequences for DNA. Therefore, it is extremely important to record the fact of interaction. Using a set of analysis methods for measuring electrical properties, it was found that the electrodes were successfully modified with lingin and the biosensor detected the presence of Mitomycin C in fish DNA. The responses of biosensors developed in previous studies were evaluated based on oxidation signals, in this study, they were assessed using the redox probe signal. This approach enabled the detection of both DNA and mitomycin C at lower concentrations compared to those reported in earlier studies.
Answer: First of all, we would like to thank the Reviewer-2 for valuable comments.
There are the following comments on the article:
- In Figure 1, it is necessary to indicate the areas of lingin modification or to indicate changes in the characteristic size of the graphite flakes. In the above form, the effect of the modification on the electron microscope images is weakly noticeable.
Answer: Thank you for your comment.
The data of SEM analysis were reviewed and Figure 1 was revised accordingly. The previous images at 5 µm scale were replaced with newly acquired images at a lower scale of 500 nm, and the relevant section has been updated. In our current study, these methods were used to show changes in surface morphology and elemental composition. However, more detailed surface analysis methods such as Raman spectroscopy or XPS are planned to be used in future studies to provide definitive evidence of lignin binding. The relevant section has been revised in Manuscript and the relevant section has been revised as follows:
‘’Microscopic characterization of PGE and PGE/LGs was carried out by the FE-SEM technique in two different dimensions, 500 nm and 1 µm (Figure 1). SEM images, despite being obtained at relatively low concentrations, revealed the accumulation of material in the form of irregular and amorphous structures in specific regions of the surface at a concentration of 750 µg/mL. These structures do not exhibit a homogeneous distribution, but rather stand out with their blotchy and irregular morphology. The observed non-uniform deposits clearly indicate that the material has interacted with the electrode surface and successfully immobilized. This suggests that the material exhibits high affinity for the surface even at low concentrations, confirming the effectiveness of the immobilization process. These findings are particularly important as the dense accumulations observed in certain areas reflect a selective adhesion tendency of the material on the electrode surface and provide insight into potential interaction mechanisms [22]’’
Figure 1. FE-SEM images of (a, c) PGE control and (b, d) PGE/LG at different (500 nm and 1 µm) magnifications with the acceleration voltage of 2.0 kV.
- On line 69-70, Mitomycin C is abbreviated by the authors as MMC, although on line 49 it is designated as MS. Subsequently, both abbreviations are present in the text, which leads to misunderstanding. It is necessary either to clarify the difference between these abbreviations, or to correct typos.
Answer: Thank you for your comment
In the article, Mitomycin C is abbreviated as MC. However, in lines 69-70, the abbreviation MMC is used, as in the reference study. This section has been revised, and the relevant section has been revised as follows:
‘’In the study conducted by Ensafi et al. [20], an impedimetric DNA biosensor was developed to evaluate the anticancer activity of the chemotherapeutic agent MC on DNA. The biosensor was constructed by immobilizing double-stranded DNA (ds-DNA) onto a pencil graphite electrode (PGE) modified with multiwalled carbon nanotubes (MWCNTs) and poly(diallyldimethylammonium chloride) (PDDA). The interaction between MC and DNA was investigated using electrochemical impedance spectroscopy (EIS). After incubating the DNA-modified electrode in MC solution for a defined time, EIS measurements were performed, and DNA damage was assessed based on changes in the charge transfer resistance (Rct). The study revealed that native MC did not exhibit significant interaction with DNA. However, upon activation either electrochemically or in acidic conditions MC demonstrated a marked interaction’’
- Why did the authors choose such a diffusion coefficient on line 205? It is necessary to add an extended description or an appropriate reference link to the source from which the number was borrowed.
Answer: Thank you for your comment
Since the electrochemical experiments were performed in a ferri/ferrocyanide redox system, the diffusion coefficient was taken as 7.6 × 10⁻⁶ cm²/s, which corresponds to the widely accepted value reported by Bard and Faulkner (2001) in their standard textbook on electrochemical methods. The reference has now been explicitly added in the manuscript to clarify the source of this parameter.
The anodic charge transfer values (Qa), anodic current values (Ia), and calculated electroactive surface areas (A) obtained for the PGE control and PGE/LG are given in Table S6. The electroactive surface areas were calculated by the Randles-Sevcik equation [29] (Eq. 1) to demonstrate that the electrode surface modified with LG in contrast to unmodified one;
Ip = (2.69 x 105) x n3/2 x A x D1/2 x C x V1/2, |
(1) |
where Ip = peak current; n = number of electrons transferred; A = electroactive surface area, cm2; D = diffusion coefficient, cm2/s (i.e, 7.6 × 10-6 cm2/s); C = concentration, mol/cm3; and V = scan rate, volt/s. [29]
- Cummings, T.E.; Elving, P.J. Determination of the Electrochemically Effective Electrode Area. Anal. Chem. 1978, 50, 480–488, doi:10.1021/ac50025a031.
- Lines 322-325 indicate the toxicity of the sample. It is desirable to expand the use of this term, since the analysis of values is based on equation (3). It describes the height of peaks before and after the redox reaction. That is, it is not entirely clear whether they are talking about the toxicity of Mitomycin C for DNA, or by toxicity the authors mean the degree of contamination of the probe.
Answer: Thank you for your comment. This section has been revised, and the relevant section has been revised as follows:
‘’As in the studies in the literature [18-19], the toxic effect of mitomycin C (MC) on DNA was assessed by examining changes in electrochemical redox peak height values before and after the interaction between MC and DNA. In this approach, the decrease in redox peak height is associated with damage or conformational changes in the DNA structure, thus indirectly determining MC binding to DNA and its toxicity. In our study, the S value calculated from the electrochemical signal obtained after incubating MC with fsDNA for 30 minutes was found to be 67%. The S value, generally defined as the proportional expression of the change in peak height, quantitatively reflects the degree of MC binding to DNA and its associated toxic effect. “The 67% value obtained indicates that MC has a moderate toxic effect on fsDNA, i.e., it causes a significant but not excessive DNA structural change. These findings support that the DNA binding mechanism of MC can be sensitively monitored by electrochemical methods and that biosensors can be used effectively in toxicity assessment. These results are consistent with previous studies reported in the literature, further validating the reliability of our findings.’’
- Findik, M.; Bingol, H.; Erdem, A. Hybrid Nanoflowers Modified Pencil Graphite Electrodes Developed for Electrochemical Monitoring of Interaction between Mitomycin C and DNA. Talanta 2021, 222, 121647, doi:10.1016/j.talanta.2020.121647.
- Erdem, A.; Muti, M.; Papakonstantinou, P.; Canavar, E.; Karadeniz, H.; Congur, G.; Sharma, S. Graphene Oxide Integrated Sensor for Electrochemical Monitoring of Mitomycin C-DNA Interaction. Analyst 2012, 137, 2129–2135, doi:10.1039/c2an16011k.
Submission Date
21 July 2025
Date of this review
01 Aug 2025 11:06:35
Reviewer 3 Report
Comments and Suggestions for Authors
This work is of interest in the field of pharmaceutical research and biomedical and environmental monitoring. The authors investigated the use of modification of graphite rods with lignin to create sensors for electrochemical detection of DNA, Mitomycin C and their interaction
However, the authors did not take into account several important issues in the work, and some points were insufficiently covered:
- Scheme 1 shows the presence of carboxyl groups in a graphite rod. It is necessary to estimate their number, as it is likely to vary from graphite rod to graphite rod. Are you sure that their number is standardized and how can you confirm this?
- Provide NHS and EDC transcripts.
- The FE-SEM and EDX methods cannot be considered informative for evaluating the binding of lignin to graphite without using additional analysis methods (for example, Raman spectroscopy, XPS). In Figure 1, there is no difference in surface roughness, so this cannot be trusted without quantitative data. The difference in oxygen by EDX is also too small to indicate the binding of lignin to graphite (lines 136-143).
- The description of the methodology is inappropriate in the discussion of the results (lines 148-153)
- All the processes of chemical bonding are described only in text, but there is no representation in the form of chemical reactions.
- Is the amount of bound lignin standardized and how reproducible are the results of crosslinking? The issue is due to the lack of data on the number of initial carboxyl groups, which casts doubt on the reproducibility of the crosslinking results.
- The alleged film formation affecting the average current value has not been proven, although it could be demonstrated at least using the SEM method (lines 185-186).
- What is the reason for the choice of toxicity limits? (lines 322-325)
In general, the work is important in the field of sensors for electrochemical detection of DNA, Mitomycin C and their interaction, but the authors did not provide sufficient evidence to confirm their work. Due to the lack of a description of chemistry in the form of reactions and schemes, it is difficult to fully understand all the stages of work, so the authors also need to describe all the alleged mechanisms of crosslinking and functioning of the sensor using reactions and drawings, confirming everything with literary data. At the moment, At the moment, the work requires additional research and elaboration in the presentation of a discussion of the results. The article can be reconsidered after major revision.
Author Response
Journal: Sensors
Manuscript ID: sensors-3801114
Title: Lignin modified single-use graphite electrodes: Electrochemical detection of DNA, Mitomycin C and their interaction
Corresponding Author: Arzum Erdem*
Author(s): Ayla Yıldırım, Meltem Maral, Huseyin Senturk, Arzum Erdem *
Received: 21 Jul 2025
August 23, 2025
The list of our answers to the comments of Editor:
Thank you for valuable comments of Editor and Reviewers. Manuscript is revised/corrected according to each comment pointed by Reviewers. The revised/corrected parts in the manuscript are shown as highlighted in red.
Reviewer 3
Open Review
( ) I would not like to sign my review report
(x) I would like to sign my review report
Quality of English Language
( ) The English could be improved to more clearly express the research.
(x) The English is fine and does not require any improvement.
Yes |
Can be improved |
Must be improved |
Not applicable |
|
Does the introduction provide sufficient background and include all relevant references? |
( ) |
(x) |
( ) |
( ) |
Is the research design appropriate? |
( ) |
( ) |
(x) |
( ) |
Are the methods adequately described? |
( ) |
(x) |
( ) |
( ) |
Are the results clearly presented? |
( ) |
( ) |
(x) |
( ) |
Are the conclusions supported by the results? |
( ) |
(x) |
( ) |
( ) |
Are all figures and tables clear and well-presented? |
(x) |
( ) |
( ) |
( ) |
Comments and Suggestions for Authors
This work is of interest in the field of pharmaceutical research and biomedical and environmental monitoring. The authors investigated the use of modification of graphite rods with lignin to create sensors for electrochemical detection of DNA, Mitomycin C and their interaction
However, the authors did not take into account several important issues in the work, and some points were insufficiently covered:
Answer: First of all, we would like to thank the Reviewer-3 for valuable comments.
- Scheme 1 shows the presence of carboxyl groups in a graphite rod. It is necessary to estimate their number, as it is likely to vary from graphite rod to graphite rod. Are you sure that their number is standardized and how can you confirm this?
Answer: Thank you for your comment
The graphite rods used in our study were obtained from commercial, identical production batches, and standard surface cleaning and activation procedures were applied beforehand to ensure consistent surface chemistry and carboxyl group density. Although SEM analysis was not performed to determine the number of graphite rods, each electrode was subjected to a pretreatment process to ensure standardization, as described in the Methods section. For each electrode, a potential of +1.2 V was applied for 30 s in ABS, and the obtained RSD values indicate the reproducibility of the results, reflecting both the uniformity of the graphite surface and the effectiveness of the electrode standardization. CV, DPV, and EIS techniques were employed in a conventional three-electrode system consisting of PGE as the working electrode, an Ag/AgCl/3 M KCl reference electrode (BAS, Model RE-5B, W. Lafayette, USA), and a platinum wire as the auxiliary electrode, with electrical contact established by soldering a metallic wire to the metallic part. The pencil was held vertically with 14 mm of the lead protruding outside, 10 mm of which was immersed in the solution. Direct quantitative measurement of the number of carboxyl groups requires evaluation using surface analysis methods such as XPS. While these analyses are limited in the current study, the reproducibility and consistency of the experimental results demonstrate that the surface properties of the graphite rods used are sufficiently standardized. In future studies, XPS analysis may be employed to investigate the surface composition and quantitatively evaluate the functional groups present on the electrode. We would like to express our sincere thanks for the valuable support.
The relevant section can be found below:
In 2.1. Procedure section
‘’To modify PGE with lignin (LG), 750 µg/mL LG solution was prepared with DMSO, and the solution was sonicated for 60 minutes.
- Preparation of LG modified PGE: Electrochemical activation of PGE surfaces was carried out by applying + 1.2 V for 30 seconds in ABS. PGEs were immersed in vi-als containing 40 µL LG solution for 30 minutes. Next, the LG-modified PGEs were dried for 15 minutes. Subsequently, the modified PGEs were activated with 5 mM EDC (1-Ethyl-3-(3-dimethylaminopropyl)carbodiimide)/8 mM NHS (N-Hydroxysuccinimide) NHS for 60 minutes. All these steps were carried out in the dark. Then the activated electrodes were washed three times with PBS.
- Immobilization of the fsDNA onto LG modified electrodes: fsDNA was immobi-lized on the PGE/LG surface for 30 minutes. Subsequently, electrodes were washed three times with ABS.
- Immobilization of the Mitomycin C (MC) onto LG modified electrodes: MC was immobilized on the PGE/LG surface for 30 minutes in the dark. Subsequently, elec-trodes were washed three times with PBS.
- Interaction of the fsDNA and MC on the electrode surface: fsDNA was immobi-lized on the PGE/LG surface for 30 minutes. Then electrodes were washed three times with ABS. Subsequently, MC was immobilized on the PGE/LG/fsDNA surface for 15, 30, and 60 minutes in the dark. Then electrodes were washed three times with PBS.’’
In Supporting Materials
“All measurements were performed in a three-electrode system in a Faraday cage. In the three-electrode system, an Ag/AgCl/3.0 M KCl electrode (BAS, Model RE-5B, W. Lafayette, USA) was used as the reference electrode, platinum wire as the auxiliary electrode, and a disposable pencil graphite electrode (PGE) as the working electrode. The pencil was held vertically with 14 mm of the lead protruding outside, 10 mm of which was immersed in the solution.”
- Provide NHS and EDC transcripts.
Answer: Thank you for your comment
The detailed information on EDC (1-Ethyl-3-(3-dimethylaminopropyl)carbodiimide) and NHS (N-Hydroxysuccinimide) chemicals and their preparation methods is provided in the supporting material section. The covalent binding solution (CA) used in this study was prepared to enable the immobilisation of biomolecules on the electrode surface. The solution was freshly prepared daily in 0.05 M phosphate-buffered saline (PBS, pH 7.4) containing 5 mM EDC and 8 mM NHS and stored at +4 °C to maintain its activity. EDC promotes covalent bonding between carboxyl groups and amine groups on the surface through its carbodiimide group, while NHS is used to enhance the efficiency of this reaction. This enhances the stability and immobilisation efficiency of biomolecules bound to the biosensor surface.
The relevant section has been revised in Supporting material section and the relevant section has been revised as follows:
‘’The covalent agent solution (CA) was prepared fresh daily with PBS (0.05 M, pH 7.4) containing 5 mM EDC (1-Ethyl-3-(3-dimethylaminopropyl)carbodiimide) /8 mM NHS (N-Hydroxysuccinimide) and kept at +4 °C.’’
The relevant section has been revised in Manuscript and the relevant section has been revised as follows:
‘’a. Preparation of LG modified PGE: Electrochemical activation of PGE surfaces was carried out by applying + 1.2 V for 30 seconds in ABS. PGEs were immersed in vi-als containing 40 µL LG solution for 30 minutes. Next, the LG-modified PGEs were dried for 15 minutes. Subsequently, the modified PGEs were activated with 5 mM EDC (1-Ethyl-3-(3-dimethylaminopropyl)carbodiimide) /8 mM NHS (N-Hydroxysuccinimide) for 60 minutes. All these steps were carried out in the dark. Then the activated electrodes were washed three times with PBS.’’
- The FE-SEM and EDX methods cannot be considered informative for evaluating the binding of lignin to graphite without using additional analysis methods (for example, Raman spectroscopy, XPS). In Figure 1, there is no difference in surface roughness, so this cannot be trusted without quantitative data. The difference in oxygen by EDX is also too small to indicate the binding of lignin to graphite (lines 136-143).
Answer: Thank you for your comment
The results of SEM analysis were reviewed and Figure 1 was revised accordingly. The previous images at 5 µm scale were replaced with newly acquired images at a lower scale of 500 nm, and the relevant section has been updated. In our current study, these methods were used to show changes in surface morphology and elemental composition. However, more detailed surface analysis methods such as Raman spectroscopy or XPS are planned to be used in future studies to provide definitive evidence of lignin binding. The relevant section has been revised in Manuscript and the relevant section has been revised as follows:
‘’Microscopic characterization of PGE and PGE/LGs was carried out by the FE-SEM technique in two different dimensions, 500 nm and 1 µm (Figure 1). SEM images, despite being obtained at relatively low concentrations, revealed the accumulation of material in the form of irregular and amorphous structures in specific regions of the surface at a concentration of 750 µg/mL. These structures do not exhibit a homogeneous distribution, but rather stand out with their blotchy and irregular morphology. The observed non-uniform deposits clearly indicate that the material has interacted with the electrode surface and successfully immobilized. This suggests that the material exhibits high affinity for the surface even at low concentrations, confirming the effectiveness of the immobilization process. These findings are particularly important as the dense accumulations observed in certain areas reflect a selective adhesion tendency of the material on the electrode surface and provide insight into potential interaction mechanisms [22]’’
Figure 1. FE-SEM images of (a, c) PGE control and (b, d) PGE/LG at different (500 nm and 1 µm) magnifications with the acceleration voltage of 2.0 kV.
- The description of the methodology is inappropriate in the discussion of the results (lines 148-153)
Answer: Thank you for your comment
The relevant section has been revised in Manuscript and the relevant section has been revised as follows:
‘’LG solutions were prepared using two different solvents (DMF and DMSO) at a concentration of 500 µg/mL and sonicated for 60 minutes. Graphite pencil electrodes (PGE) were modified by immersing them in these LG solutions for 30 minutes, then dried at room temperature for 15 minutes. Following modification, the electrodes were activated with a covalent coupling agent solution (CA) for 60 minutes to ensure strong and stable binding of biomolecules to the surface. The EDC in this solution promotes covalent bonding between the carboxyl groups and amine groups on the electrode surface via a carbodiimide group, while NHS increases the efficiency of this reaction, ensuring more effective bonding. As a result, the stability and immobilisation efficiency of biomolecules immobilised on the biosensor surface have been significantly increased [23]’’
- All the processes of chemical bonding are described only in text, but there is no representation in the form of chemical reactions.
Answer: Thank you for your comment
‘’The NHS contributes positively to this process by increasing the stability of EDC [24]. To ensure efficient immobilization of DNA on the modified electrode surface, all subsequent experiments were conducted in the presence of the EDC/NHS activation step. Although lignin is inherently a non-conductive material, it contains abundant hydroxyl, methoxy, and phenolic functional groups in its structure [25]. These groups can form strong interactions with biomolecules such as DNA or proteins [26]. These interactions can occur through covalent bonding (e.g., EDC/NHS coupling chemistry) or non-covalent forces such as hydrogen bonds and π–π interactions. This functionally rich surface enhances the efficiency and stability of biomolecule immobilization on the electrode surface. Improved immobilization increases the surface density of the recognition elements, enabling more efficient capture of target molecules and reducing sig-nal loss. Despite of the lack of conductivity due to lignin immobilization, the overall detection sensitivity of the PEG/LG composite electrode is enhanced [27,28].’’
Added references:
- Laurichesse, S.; Avérous, L. Chemical Modification of Lignins: Towards Biobased Polymers. Prog. Polym. Sci. 2014, 39, 1266–1290.
- Zhao, J.; Zhu, M.; Jin, W.; Zhang, J.; Fan, G.; Feng, Y.; Li, Z.; Wang, S.; Lee, J.S.; Luan, G.; et al. A Comprehensive Review of Unlocking the Potential of Lignin-Derived Biomaterials: From Lignin Structure to Biomedical Application. J. Nanobiotechnology 2025, 23.
- Tunca, N.; Maral, M.; Yildiz, E.; Sengel, S.B.; Erdem, A. Synthesis and Characterization of Polysaccharide-Cryogel and Its Application to the Electrochemical Detection of DNA. Microchim. Acta 2024, 191, doi:10.1007/s00604-024-06550-7.
- Tortolini, C.; Capecchi, E.; Tasca, F.; Pofi, R.; Venneri, M.A.; Saladino, R.; Antiochia, R. Novel Nanoarchitectures Based on Lignin Nanoparticles for Electrochemical Eco‐friendly Biosensing Development. Nanomaterials 2021, 11, 1–17, doi:10.3390/nano11030718.
- Is the amount of bound lignin standardized and how reproducible are the results of crosslinking? The issue is due to the lack of data on the number of initial carboxyl groups, which casts doubt on the reproducibility of the crosslinking results.
Answer: Thank you for your comment
In our study, lignin solutions were prepared at constant concentrations to standardize the lignin content and, consequently, the carboxyl group density, while modification conditions were carefully controlled. The crosslinking process was repeated in each experiment, and the consistency and reproducibility of the results were evaluated using electrochemical measurements. Changes in the average current values of the electrodes before and after lignin binding, as well as before and after crosslinking, along with their reproducibility data, are provided in the supplementary materials (Table S4). The reported current values represent the average of three independent measurements (n = 3). Advanced surface analyses, such as XPS, for direct quantification of initial carboxyl group density are planned for future studies. The relevant Supporting material section can be found below:
Table S4. Average current values (n = 3) obtained by CV after EDC/NHS activation of LG modified electrodes for different times.
CA activation time (min) |
Electrodes |
Ia (µA) |
RSD (%) |
% Change in current value* |
0.28 V |
||||
5 |
PGE |
243.25 ± 4.40 |
1.81 |
- |
PGE/LG |
264.37 ± 3.00 |
1.13 |
9% increase |
|
30 |
PGE |
236.90 ± 16.18 |
6.83 |
- |
PGE/LG |
263.23 ± 7.46 |
2.83 |
11% increase |
|
60 |
PGE |
242.23 ± 4.61 |
1.90 |
- |
PGE/LG |
291.24 ± 2.03 |
0.70 |
20% increase |
*% change values were calculated based on the average current values (n = 3) obtained with control groups (PGE).
- The alleged film formation affecting the average current value has not been proven, although it could be demonstrated at least using the SEM method (lines 185-186).
Answer: Thank you for your comment
Our experimental electrochemical results indicate that as lignin concentration increases above the optimum value, a progressively thicker film layer forms on the electrode surface. This thickened lignin layer acts as a physical barrier in the electron transfer process between the electrode surface and the redox-active species in solution. As the film thickens, the distance electrons must travel to reach the electrode surface increases, resulting in increased charge transfer resistance. Furthermore, a thick film reduces the surface's electroactive area, thus narrowing the active region where electrochemical reactions occur. These effects, combined, lead to a slowdown in electron transfer kinetics and a decrease in the current signal observed in electrochemical measurements. Therefore, controlling lignin film thickness is crucial to ensure adequate lignin immobilization while not negatively impacting electron transfer efficiency at the electrode surface. The relevant section has been revised in Manuscript and the relevant section has been revised as follows:
‘’To study the effect of LG concentration on surface modification, LG solutions were prepared at six different concentrations, and PGE surfaces were modified. After 250, 500, 750, 1000, 1500, and 3000 µg/mL LG modification onto PGE, there was an increase in average current value by 3%, 17%, 20%, 15%, 14%, and 9%, respectively, compared to the control group. The oxidation current increased progressively with increasing LG concentration, reaching a maximum enhancement of 20% at 750 µg/mL compared to the unmodified control. In addition, a proportional decrease in current was observed with increasing LG concentration (1000 to 3000 µg/mL), likely due to the formation of a thicker film on the electrode surface that limited electron transfer. Therefore, the optimum LG concentration was determined as 750 µg/mL (Figure 2B, Table S2).”
- What is the reason for the choice of toxicity limits? (lines 322-325)
Answer: Thank you for your comment
‘’As in the studies in the literature [18-19], the toxic effect of mitomycin C (MC) on DNA was assessed by examining changes in electrochemical redox peak height values before and after the interaction between MC and DNA. In this approach, the decrease in redox peak height is associated with damage or conformational changes in the DNA structure, thus indirectly determining MC binding to DNA and its toxicity. In our study, the S value calculated from the electrochemical signal obtained after incubating MC with fsDNA for 30 minutes was found to be 67%. The S value, generally defined as the proportional expression of the change in peak height, quantitatively reflects the degree of MC binding to DNA and its associated toxic effect. “The 67% value obtained indicates that MC has a moderate toxic effect on fsDNA, i.e., it causes a significant but not excessive DNA structural change. These findings support that the DNA binding mechanism of MC can be sensitively monitored by electrochemical methods and that biosensors can be used effectively in toxicity assessment. These results are consistent with previous studies reported in the literature, further validating the reliability of our findings.’’
- Findik, M.; Bingol, H.; Erdem, A. Hybrid Nanoflowers Modified Pencil Graphite Electrodes Developed for Electrochemical Monitoring of Interaction between Mitomycin C and DNA. Talanta 2021, 222, 121647, doi:10.1016/j.talanta.2020.121647.
- Erdem, A.; Muti, M.; Papakonstantinou, P.; Canavar, E.; Karadeniz, H.; Congur, G.; Sharma, S. Graphene Oxide Integrated Sensor for Electrochemical Monitoring of Mitomycin C-DNA Interaction. Analyst 2012, 137, 2129–2135, doi:10.1039/c2an16011k.
In general, the work is important in the field of sensors for electrochemical detection of DNA, Mitomycin C and their interaction, but the authors did not provide sufficient evidence to confirm their work. Due to the lack of a description of chemistry in the form of reactions and schemes, it is difficult to fully understand all the stages of work, so the authors also need to describe all the alleged mechanisms of crosslinking and functioning of the sensor using reactions and drawings, confirming everything with literary data. At the moment, At the moment, the work requires additional research and elaboration in the presentation of a discussion of the results. The article can be reconsidered after major revision.
Submission Date
21 July 2025
Date of this review
04 Aug 2025 12:28:2
Round 2
Reviewer 1 Report
Comments and Suggestions for Authors
I recommend the acceptance of this manuscript as it is.
Reviewer 3 Report
Comments and Suggestions for Authors
The authors have corrected all comments.